# Color Stability of Dental Reinforced CAD/CAM Hybrid Composite Blocks Compared to Regular Blocks

**DOI:** 10.3390/ma13214722

**Published:** 2020-10-22

**Authors:** Yeong-Ah Kang, Han-Ah Lee, Joseph Chang, Wonjoon Moon, Shin Hye Chung, Bum-Soon Lim

**Affiliations:** Department of Dental Biomaterials Science, School of Dentistry and Dental Research Institute, Seoul National University, Seoul 03080, Korea; yeongah1984@snu.ac.kr (Y.-A.K.); han6936@snu.ac.kr (H.-A.L.); ssebjomanji@snu.ac.kr (J.C.); w.moon@snu.ac.kr (W.M.); den533@snu.ac.kr (S.H.C.)

**Keywords:** CAD/CAM hybrid composite block, reinforced and regular block, color stability, translucency, simulated red wine, microstructural feature

## Abstract

This study compares the color stability of dental reinforced computer-aided design/computer-aided manufacturing (CAD/CAM) hybrid composite blocks to that of regular blocks. One hundred fifty disc-type specimens (n = 15) were prepared from five sets of hybrid composite blocks (Cerasmart-200/Cerasmart-300, KZR-CAD HR/KZR-CAD HR3, Estelite Block/ Estelite-P Block, Avencia Block/Avencia-P Block, Mazic Duro/Duro Ace). The specimen color and translucency parameter (TP) were assessed using a spectrophotometer before and after immersion in staining solutions (water, 10% ethanol, simulated red wine). Changes in color (ΔE) and translucency (ΔTP) of specimens were calculated. The data were analyzed using the analysis of variance (ANOVA) and Tukey’s post hoc test (*p* < 0.05). Microstructural features of the hybrid composite blocks were also examined using FE-SEM. Immersion in deionized water or 10% ethanol made no significant color or translucency changes (except for Avencia-P Block); however, the simulated red wine caused significant changes to the color and translucency of almost all specimens, especially after 4 weeks of immersion. The reinforced hybrid blocks (except for Estelite-P Block and Duro Ace) showed lower color stability than corresponding regular blocks. Avencia-P Block showed significantly reduced color stability compared to Avencia Block. Even in deionized water and 10% ethanol, Avencia-P Block showed perceptible ΔE and decreased translucency. Estelite Block/ Estelite-P Block and Mazic Duro/Duro Ace showed better color stability than the other materials tested.

## 1. Introduction

In dentistry, the use of computer-aided design/computer-aided manufacturing (CAD/CAM) has rapidly increased in recent years due to dramatic technological advances [1]. New classes of CAD/CAM materials, such as hybrid composite resins, have been introduced specifically for esthetic restorations. These materials overcome the disadvantages associated with the use of direct composite resins in clinical dentistry with their standardized manufacturing process and the lack of polymerization defects induced during their application [2].

Over the years, the mechanical properties of CAD/CAM hybrid composite blocks have improved through the alteration of the resin matrix and the incorporation of filler particles [3,4,5]. Hybrid composite blocks have different microstructures as well as variable filler contents and hence differences in their mechanical properties. Recently, reinforced CAD/CAM hybrid composite blocks were developed to restore the posterior teeth, which are subject to high masticatory forces. For example, Cerasmart 300 (GC, Japan) was introduced as the reinforced hybrid composite block for Cerasmart 200. 

There are few studies on mechanical properties of the reinforced dental hybrid composite blocks [6]. In addition to the initial color match and translucency of the restoration, the clinical esthetic stability throughout the functional lifetime is as important as the mechanical properties of the reinforced hybrid composite block [7,8,9]. Color or translucency changes over time may limit the longevity and quality of esthetic restorations. Certain CAD/CAM restorative materials may change color when subjected to staining solutions that simulate the consumption of regular beverages [10,11,12].

When exposed to an oral environment, discoloration of hybrid composite blocks can occur due to extrinsic and intrinsic factors. Extrinsic factors include staining by adsorption or absorption of colorants from food and beverages. Intrinsic stains may be due to an alteration of the material itself (such as types of resin matrix, filler particles, photo-initiator system). The color changes of dental esthetic materials can be visually assessed, but can be assessed more quantitatively using analytical equipment such as a spectrophotometer and spectroradiometer. Spectrophotometry with CIE L*a*b* systems is a commonly used method to quantitatively measure color and translucency in dentistry [13].

During color measurements, both the actual color of the surface and the surface’s lighting conditions will affect the readings. A white background has been used as a standard background, but in a recent study, a black background was considered a more appropriate background for anterior teeth. The black background is intended to simulate the light reflectance in restorations that are not surrounded by cavity walls, and the white background is intended to simulate the light reflectance in restorations that are surrounded by tooth walls [9,11,14]. 

The translucency of dental esthetic restorations is mainly evaluated by the translucency parameter (TP) or contrast ratio (CR). In this study, the TP was evaluated by measuring the color coefficients on a white and black background. Translucency has been emphasized as one of the primary factors in controlling the esthetic outcome because it makes esthetic restorations appear more natural [15,16]. However, little information is available regarding the translucent characteristics of contemporary CAD/CAM hybrid composite blocks. Due to different translucencies, the esthetic properties of restorations can vary greatly, so some manufacturers produce and release different translucencies (LT, HT) of the same shade blocks. 

Food and beverages that are primarily used to assess the color stability of esthetic restorations include coffee, wine, tea, cola, and curry. It has been reported that colored beverages, particularly red wine, have a significant effect on the color of the esthetic restorations [8,9,10,11,17]. Although the pigment (anthocyanin) in red wine is considered to have an effect on discoloration, it has been difficult to quantitatively compare the effects of anthocyanin because commercially available red wines, which contain various components (alcohol, acids, chromogens, and tannins), were used in most discoloration tests. In this study, the simulated red wine was prepared directly with anthocyanin and ethanol (alcohol) to conduct a more quantitative evaluation. Alcohol is also a suspected factor that affects the color stability of the esthetic restorations. Alcohol-containing mouthwash solutions have also been found to cause more discoloration than non-alcoholic mouthwash solutions. There have also been reports of noticeable discoloration in resins exposed to alcoholic beverages such as beer and whiskey [18,19,20].

CAD/CAM hybrid composite blocks have generally acceptable esthetic properties but insufficient strength, which limits their applications in the areas of high occlusal stresses. Reinforced hybrid composite blocks have been introduced to provide improved mechanical strength without sacrificing the esthetic characteristics of the restoration. However, few studies have been found to assess the potential change in the color and translucency of the reinforced hybrid composite blocks in the oral environment. The aim of this study is to evaluate changes to the color and translucency of currently available reinforced CAD/CAM hybrid composite blocks after exposure to different staining solutions. The tested null hypothesis was that the color and translucency of the reinforced CAD/CAM hybrid composite blocks would be equal to or less susceptible than the corresponding regular blocks upon immersion in different staining solutions.

## 2. Materials and Methods

### 2.1. Hybrid Composite Block Specimen and Staining Solution

Ten CAD/CAM hybrid composite blocks produced by five different manufacturers were selected. The main components, filler fractions, and their manufacturers are listed in Table 1.

A total of 150 specimens were prepared from the 10 tested hybrid composite block groups (n = 15). The specimens from each group were divided into three subgroups (n = 5) that were assigned to different staining solutions. Fifteen disk type specimens (10 × 10 × 2 mm) were prepared by cutting each hybrid composite using a low speed diamond saw (Isomet 1000, Buehler, USA). The surfaces of all specimens were polished with SiC paper (#800, #1200, and #2000). The thickness of all specimens was confirmed to be 2.0 ± 0.01 mm using a digital micrometer. All specimens were then ultrasonically cleaned in deionized water for 5 min.

The specimens from each tested block were randomly divided into three subgroups (n = 5) for the three staining solutions (deionized water, 10% ethanol, and simulated red wine). The 10% ethanol solution was prepared by pouring 100 mL ethanol (99.5%, Samchun, Korea) into 900 mL deionized water, after which the solution was stirred for 30 min. To prepare the simulated red wine solution, 25 mg Kuromanin chloride (CAS no. 7084-24-4; Chengdu Biopurity Phytochemical, China), which is used as an edible red pigment, was poured into 500 mL 10% ethanol solution, and the solution was stirred for 1 h.

### 2.2. Color and Translucency Measurement

The specimen color and translucency were measured using a reflective spectrophotometer (CiXX0, X-rite, USA). The aperture size was set to 6 mm and the specimens were positioned in the center of the measuring port. The color coordinate (CIE L*a*b*) of the specimens was measured using a spectrophotometer against white and black backgrounds relative to the CIE standard illuminant D65. The spectrophotometer was calibrated before each measurement. Three measurements were taken for each specimen and the average was recorded.

The specimens were immersed individually in closed vials containing 5 mL of the staining solution and stored in a shaking incubator at 37 °C for up to 12 weeks. The solutions were freshened weekly to avoid degradation. At specific times of the immersion period, the specimens were lightly rinsed with deionized water and wiped with the tissue (Kimwipes, Korea) for color and translucency measurements. Then, the specimens were reimmersed in the staining solutions for further treatment.

The specimen color was assessed after employing different staining protocols as described for the baseline measurements. The color change (ΔE) of each specimen was calculated using the following formula:ΔE*^*^* = [(*L*_after staining_ − *L*_baseline_)^2^ + (*a*_after staining_ − *a*_baseline_)^2^ + (*b*_after staining_ − *b*_baseline_)^2^]^0.5^

The translucency parameter (TP) values were obtained by calculating the difference in the color of the specimens against black and white backgrounds using the following formula:TP = [(*L^*^*_B_ − *L^*^*_W_)^2^ + (*a^*^*_B_ − *a^*^*_W_)^2^ + (*b^*^*_B_ − *b^*^*_W_)^2^]^0.5^
where the subscripts B and W refer to the color coordinates against the black and white backgrounds, respectively.

Differences in the TP values were calculated using the following formula:ΔTP = TP_after staining_ − TP_baseline_

The mean and standard deviation values of ΔE and ΔTP were calculated for each subgroup. The data were analyzed using the analysis of variance (ANOVA) and Tukey’s post hoc test (*p* < 0.05).

### 2.3. Examination of Microstructure of Hybrid Block

Specimens were prepared for examination of the microstructural features of the hybrid composite blocks. Each specimen (1 mm thick) was immersed in a 30% HF solution for 30 min to dissolve silica and silicate filler particles. Then, the etched specimens were sonically cleaned in a deionized water bath for 30 min, and then in an isopropyl alcohol bath for additional 5 min before they were gold coated and examined under the SEM (S-4700, Hitachi, Japan) for the qualitative analyses. Images at two different magnifications (10,000× and 20,000×) were recorded.

## 3. Results

The means and standard deviations of the color changes (ΔE) are presented in Table 2. In some hybrid blocks, the color changes were visually discernible. ΔE of CER2 in deionized water was observed to be greater than that of CER3, but the difference was not significant (*p* > 0.05), and the color change was not visually perceptible. Similarly, CER3 showed more color change than CER2 in 10% ethanol, but not to the extent that it could be visually distinguished. Specimens from 4 weeks of immersion in the simulated red wine showed significant color changes (ΔE > 3.0); color changes of CER3 were observed to be significantly greater than those of CER2 (*p* < 0.05). CER2 showed a higher translucency parameter (TP) before immersion in the staining solution than CER3 (Table 3). The TP of CER2 in deionized water and 10% ethanol initially showed no significant change, but after 4 weeks, TP decreased gradually. Likewise, CER3 maintained its TP without significant changes in deionized water and 10% ethanol, and the ΔTP showed signs of gradual increase in the simulated red wine (*p* < 0.05).

The KZR showed a similar behavior of color change to CER. Color changes of the reinforced block (KZR3) was greater than that of the regular block (KZR). KZR showed rapid color changes after 4 weeks of immersion in the simulated red wine and slightly decreased over time. KZR3 showed a gradual increase in ΔE in the simulated red wine (Figure 1). The initial TP measured before immersion was similar for both materials (KZR and KZR3) and showed no significant difference from the initial TP level upon immersion in deionized water and 10% ethanol. In the simulated red wine, TP decreased significantly with increasing immersion time, and ΔTP was higher for KZR3 than for KZR.

The trend of ΔE in EST (regular block) was generally similar to that in CER and KZR. ΔE of the ESTP (reinforced block) was observed to be significantly greater than that of the EST (regular), but the overall color changes were less than those of CER and KZR. The EST also showed rapid color changes at 4 weeks after the immersion in the simulated red wine. Significant ΔTPs were not observed in the specimens immersed in deionized water and 10% ethanol, but in the case of the simulated red wine, the TP was significantly reduced with increasing immersion time, and the ΔTP in the ESTP was observed to be greater than that in the EST.

Unlike the other materials, AVEP (reinforced block) started color changes in deionized water, and the color change was perceptible (ΔE > 3.0) after 2 weeks of immersion. Compared to AVE (regular block), AVEP was more sensitive to 10% ethanol, resulting in higher ΔEs. In the simulated red wine, the AVEP showed nearly 5 times as large ΔE as the AVE. Notably, AVE showed very small ΔEs compared to other groups even 8 weeks after immersion in the simulated red wine, demonstrating excellent color stability. The initial TPs of the AVE and the AVEP were similar and did not change significantly with immersion in deionized water or 10% ethanol, even after extended immersion times. However, the specimens immersed in the simulated red wine showed a significant decrease in TP with increasing immersion time (Table 3), while the AVE had a lower degree of ΔTP than the others.

The DRA is the only material that exhibited less ΔE for the reinforced block. A small noticeable change in color was only observed from this specimen after immersion in the simulated red wine; there was no significant change in color after immersion in deionized water or 10% ethanol, even with an extended immersion time. Both MAZ and DRA maintained almost constant TPs in deionized water and 10% ethanol, but their TPs showed a significant decrease in the simulated red wine as the immersion time increased. The ΔTP of MAZ was more than that of DRA.

Figure 2 shows SEM images of hybrid composite block (regular, reinforced) specimens treated in a 30% HF solution. Obvious microstructure differences can be observed between the regular and reinforced blocks. Comparing CER2 and CER3, the overall filler size of the reinforced block (CER3) was larger than that of the regular block (CER2). In the case of KZR and KZR3, it could be observed that larger-sized filler particles were added to the reinforced block (KZR3). For EST and ESTP, it could be seen that a small amount of large filler particles was added in the reinforced block. In the case of AVE and AVEP, the filler particles of the reinforced block and the filler particles of the regular block were totally different. In the case of MAZ and DRA, large filler particles were also observed in the reinforced block (DRA), and the round particles shown in the image are presumed to be zirconia particles remaining on the surface that were not dissolved during hydrofluoric acid treatment.

## 4. Discussion

CAD/CAM hybrid composite block is a material widely used in dental esthetic restoration. While the reinforced hybrid composite blocks were studied for their potential for improving the mechanical properties to withstand high biting forces, the loss of color stability of the esthetic restorations could be a serious problem. In the preliminary flexural strength test in this laboratory, the flexural strength of Cerasmart 300 (217.2 MPa) was increased by 13.4% from Cerasmart 200 (191.6 MPa), and the flexural strength of the other reinforced blocks was measured at an increase of 21.1–58.6% over the regular blocks (unpublished data). Maintaining color and translucency stability in all esthetic restorations is an important factor in the success of the treatments [13,21]. The esthetic restorations are always at risk of discoloration in the oral environment, as they come into contact with various beverages and foods. Therefore, evaluation of color stability is essential, even in the reinforced hybrid composite blocks. In this study, the color and translucency changes of regular and reinforced hybrid composite blocks were evaluated. The tested reinforced hybrid blocks (except ESTP and DRA) showed lower color stability than their regular hybrid block counterparts. The null hypothesis was partially rejected.

Various methods are being used in dentistry to evaluate color changes (ΔE). The degree of discoloration may be determined visually, but more accurate, reproducible, and objective assessments can be obtained through quantification using analytical equipment [13,22]. In this study, the CIE L*a*b* color coefficient was measured using a spectrophotometer. The ΔE of specimen resulting from immersion in the various staining solutions were calculated. Generally, a value of ΔE greater than 1.0 is known to be visually recognizable, and a value greater than 3.3 is considered clinically unacceptable [23,24]. The high standard deviation may be due to the formation of non-homogenous discoloration in the specimen, and the measuring port (diameter, 6 mm) could not be placed in the exact same region.

When measuring the L*a*b* color coordinates of a specimen in a spectrophotometer, the specimen can be measured without a background or with a colored acrylic plate background, such as white or black. In some cases, a negative gray located at the mid-value on the L axis may be tested against a background of a negative gray because it has a minimal effect on the measurement of the lightness [10]. While color measurements are predominantly performed using white backgrounds, recent studies have suggested that color measurements using black backgrounds are more ideal for reproducing oral environments. A white background reproduces a condition in which the restoration is placed on tooth tissue, whereas a black background reproduces the conditions of dark oral environments [9,11,14]. It has been reported that the measured color change is more noticeable when the CAD/CAM composite resin block is measured using a white background [17]. In this study, ΔE and ΔTP were calculated by measuring color coordinates using both white and black backgrounds.

The esthetic properties of dental restorations are affected by color and color components such as shade, lightness, and chroma, as well as translucency. When visible light is irradiated on natural teeth or restorations, some reflects off the surface, some disperses, and the rest of the light passes through. The reflected or passed light reaches the eye and enters the brain as visual information which is recognized as the color of the teeth. In addition, translucency, opacity, milky light, and fluorescence make up the secondary optical properties of natural color values. Furthermore, depending on the various light sources used, the recognized tone of the esthetic restoration may vary significantly [25,26,27]. In this study, D65 light sources were used to measure color coordinates, which are standardized CIE standard light sources that are close to daylight in natural conditions outdoors, including ultraviolet light.

The factors causing discoloration of natural teeth and esthetic restorations can be divided into internal and external factors. In this study, an external factor—i.e., a staining solution—was simulated to evaluate the color stabilities. Coffee, cola, red wine, and tea have been primarily used as staining solutions in research that studies the effects of beverages on the color stability of esthetic restorations. Most of these studies found that red wine was the most staining beverage [7,17,28,29]. Red wine has been reported to have an effect on the discoloration of restorations because it contains phenolic compounds, such as tannin or anthocyanin [13]. In this study, experiments were designed using red wine to compare the color stability of regular hybrid composite blocks with that of reinforced blocks. Most studies using red wine have utilized commercial red wine that may contain variations in ethanol, pH, tannin, and red pigment that can affect discolorations. Therefore, in this study, a simulated red wine was prepared and used to simplify the number of components to interrogate the effects of ethanol and red pigment alone. Most commercial wines contain between 12 and 13.5% alcohol, and in this study, 10% ethanol solution and standard red pigment (kuromanin chloride) were mixed to prepare the simulated red wine solutions. The color change by red wine results from a complex process caused by the degradation of the resin matrix by ethanol and acid and surface deposition of pigments, which can further trigger discoloration by increasing the adsorption of pigments to the surface of the restoration [17].

Ethanol is a bipolar molecule that can promote solubility of hydrophilic and hydrophobic components, and many studies have shown that ethanol can soften and decompose the resin matrix [21,30,31]. In this study, clinically detectable levels of ΔE were observed in the specimens after 8 weeks of immersion in colorless 10% ethanol solution. However, the observed ΔEs in 10% ethanol were similar to those in deionized water, which is in agreement with previous reports [13]. It has been reported that clinically perceptible ΔEs were observed after 4 weeks when CAD/CAM composite resin blocks were exposed to 2.5% anthocyanin solution without ethanol, whereas perceptible changes were observed after immersion for 3 weeks in 12.5% [32]. On the other hand, 2.5% anthocyanin solutions containing 40% ethanol have been reported to cause significant ΔEs after 3 weeks and 12.5% anthocyanin solutions after 1 week. These differences may have been influenced by the differences in concentration of ethanol, type of wine, immersion times, etc. In this study, most specimens that were immersed in 10% ethanol showed clinically acceptable ΔE values (ΔE > 3.0); however, most specimens immersed in the simulated red wine showed clinically unacceptable ΔE (ΔE < 3.0).

To improve the properties of CAD/CAD hybrid composite blocks, changes to the resin matrix and filler particles can be independently considered. Such changes may include the composition of the resin matrix, the ratio of the existing matrix composition, the composition and fraction of the filler particle, and the particles’ sizes and shapes. It has been reported that the strength and elastic modulus of CAD/CAM hybrid composite blocks are related to the filler fractions [3]. Of the five sets of hybrid composite blocks evaluated in this study, all the reinforced blocks were observed to have larger filler fractions than their corresponding regular blocks, except for the KZR. However, the color stability of the reinforced block was not improved compared to the regular block. In particular, AVEP (reinforced block) was observed to have significantly reduced color stability, even though the filler fraction increased by 20%, compared to AVE (regular block). It has been reported that the resin matrix appears to have more influence on color changes than the filler fraction [33].

Results from previous studies have shown that the fraction of resin matrix is not directly correlated to ΔE and that the characteristics of the resin monomer play more significant roles than the fraction of the resin matrix [10,29,34,35]. Color stability of composite resins can be greatly influenced by water absorption and the hydrophilicity of the resin matrix, and that the total amount of water absorbed can be greatly affected by the level of coupling between the filler and resin matrix [13]. Composite resin is known to allow water to penetrate the substrate or the filler-resin interface [36]. If composite resins can absorb water, then water-soluble pigments can also be absorbed, increasing the ΔE. Hydrophilic resins are found to be more water-absorbing and more sensitive to water-soluble coloring solutions than hydrophobic resins [24,37].

Bis-GMA absorbs more water than UDMA, TEGDMA, and Bis-EMA, while Bis-GMA appears to induce hydrophilic pigmentation by absorbing relatively large amounts of water [38]. UDMA, which does not include a hydroxyl side group, is less hydrophilic, viscous, and soluble than Bis-GMA, resulting in a high level of color stability [22,39,40]. In addition, most composite resin manufacturers use UDMA rather than Bis-GMA due to the biocompatibility of BPA [5,41], and UDMA is a major resin component in all the hybrid composite blocks used in this study, except for MAZ. In this study, improved color stability was only observed in the reinforced type of DRA, which replaces Bis-GMA with UDMA and increases the content of ceramic filler compared to MAZ (regular block). However, further research is needed to determine whether enhanced color stability is more affected by changes to the resin matrix or to the filler particles.

Studies have shown that the treatment of silane on the filler particle plays an important role in color stability of composite resins, as much as the type of resin matrix, since ceramic filler particles can be stained at the interface when filler–resin matrix binding is not sufficient [14,29,42]. In this respect, the silane treatment of fillers is important for securing long-term color stability of composite resins. Previous studies have shown that the extremely fine space between the filler and resin substrate is suspected to provide a residence or penetration passage for coloring ingredients [43,44]. It has been reported that proper silane treatment of nanofiller particles may be a difficult process [45]. Thus, the differences in the effect of silane treatment of nanofillers added to the CAD/CAM hybrid composite block or the effect of silane treatment on various filler particles may strongly affect the color stability of the hybrid composite block.

Although gaps between nanofillers and the resin matrix were difficult to eliminate, nanofillers could be impregnated into the resin matrix and bind tightly by polymerization. Increasing the size of the filler particle makes it difficult to tightly bind and form microgaps at the interfaces between the filler and resin matrix. This may result in a partial break of the resin and filler coupling, which can have a fatal effect on the color stability of the hybrid composite block. In the SEM photographs of the specimens observed in this study, the sizes and shapes of the fillers used in each hybrid composite block showed great differences, and the binding strength and binding durability of these filler particles and the resin matrix seem to have played an important role in the color stability of each hybrid composite block.

SEM images observing the surface of the hybrid composite block showed that the reinforced blocks consisted of filler particles larger in size than the regular blocks, so it could be suggested that the particle size is correlated with the color stability of hybrid composite blocks. It has also been suggested that the size of the filler particles contained in the composite and the properties of the resin monomer are more closely related to the discoloration of CAD/CAM blocks than the filler content [46]. Although the main components released by the 3M ESPE are similar, conflicting discoloration results were published in studies evaluated after the two composite resins (Filtek Supreme, Filtek Z250) of different filler sizes were immersed in a 24-h test solution. Better color stability was observed from the Filtek Z250 microhybrid composite resins in one study [11], whereas another study found improved color stability in the Filtek Supreme nanohybrid composite resins [29]. Another research group, who conducted similar research with the same product, reported that the nanofilled composite resin (Filtek Z350 XT) shows better color stability than the microhybrid composite resin (Filtek Z250), which makes it difficult to suggest a direct correlation between filler size and color stability [19].

When considering the optical properties of dental esthetic restorations, translucency is an important factor in satisfying esthetic properties The translucency may not only depend on the thickness and surface roughness of the block, but also on the composition of the hybrid composite block. Translucency is mainly affected by differences in refractive index between the filler and resin matrix, filler size, and fraction [15,47]. Increasing the size of filler particles can increase light absorption and cause multi-directional light dispersion within the hybrid composite blocks. The more translucent hybrid blocks showed better color stability than the less translucent hybrid blocks when immersed in red wine [16,48].

ΔTP values greater than two are considered perceivable [49]. In this study, the translucencies of the specimens were significantly reduced after 4 weeks of immersion in the simulated red wine (except for AVE/AVEP and DRA). The AVE/AVEP group showed less ΔTP than the other groups, which is very unusual compared to the higher ΔE. These results could suggest that the filler fraction is not directly correlated with the translucency. It was reported that CAD/CAM composite resin blocks immersed in red wine for 1 week showed large ΔTP values (2.01–3.87) [17]. In this study, however, there was no significant change of TP after 1 week of immersion in red wine. The observed differences in ΔTP may be due to differences in testing methods, such as differences between commercial and experimental red wines. CER2/CER3, AVEP, and MAZ/DRA groups contain barium glass as fillers. The translucency parameter of CER2/CER3 was decreased with increasing immersion time; however, the TPs of AVEP and DRA were not significantly changed over the entire immersion period. Thus, rather than translucency being solely determined by filler type or fraction, various factors such as filler size, resin monomer, silane treatment, and manufacturing process in combination are thought to determine the translucency stability.

## 5. Conclusions

Clinical esthetic stability of CAD/CAM hybrid composite blocks throughout their functional lifetimes as well as the initial color match and translucency of a restoration are as important as the mechanical properties of the reinforced hybrid composite block. The tested reinforced hybrid blocks (except DRA and ESTP) showed lower color stability than their regular hybrid block counterparts. EST/ESTP and MAZ/DRA showed better stain resistance than the others. Where both esthetic and strong mechanical properties are required, care should be taken to apply the hybrid composite blocks. Further study of the correlation between the microstructural features and flexural strength of hybrid block are necessary to improve our understanding of these materials.

## Figures and Tables

**Figure 1 materials-13-04722-f001:**
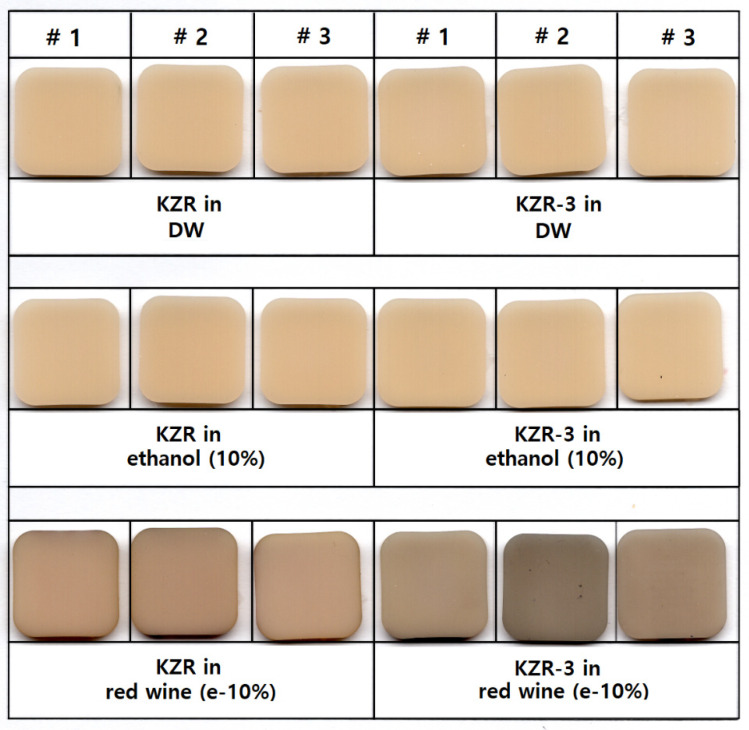
KZR and KZR-3 block specimens after color stability test (12-week immersion in staining solutions).

**Figure 2 materials-13-04722-f002:**
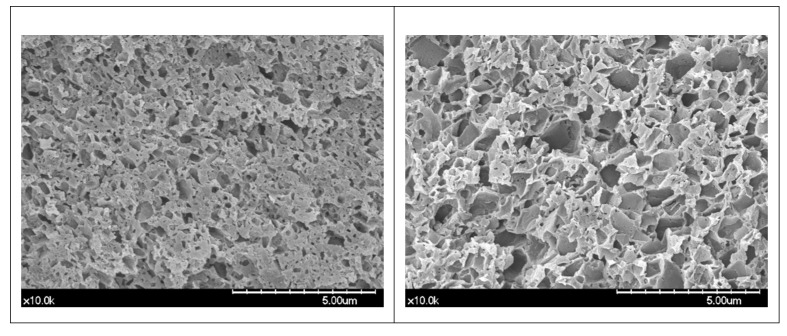
Microstructures of hybrid composite blocks after 30% HF solution treatment (top = original magnification 10k, bottom = original magnification 20k, left = the regular block, right = the reinforced block).

**Table 1 materials-13-04722-t001:** Computer-aided design/computer-aided manufacturing (CAD/CAM) hybrid composite resin blocks used in this study.

Code	Name	Main Components	Filler Fraction	Lot #	Manufacturer
CER2	Cerasmart 200	Bis-MEPP *, UDMA, DMA, Ba-glass, silica	71 wt.%	1603081	GC, Japan
CER3	Cerasmart 300	UDMA, Bis-MEPP, Ba-glass, silica	78 wt.%	1808292	
KZR	KZR-CAD HR	UDMA, TEGDMA, SiO_2_ + Al_2_O_3_ + ZrO_2_, SiO_2_	79 wt.%	01011622	Yamakin, Japan
KZR3	KZR-CAD HR3	UDMA, DEGDMA, SiO_2_ + Al_2_O_3_ + ZrO_2_, SiO_2_	75 wt.%	01111807	
EST	Estelite Block	UDMA, TEGDMA, silica, silica-zirconia	75 wt.%	0070Z5	Tokuyama Dental, Japan
ESTP	Estelite P Block	Bis-MPEPP, UDMA, NPGDMA, silica, silica-zirconia filler	81 wt.%	011049	
AVE	Katana Avencia Block	UDMA, TEGDMA, silica, alumina filler	62 wt.%	000123	Kuraray, Japan
AVEP	Katana Avencia P Block	UDMA, Ba-glass, silica	82 wt.%	000043	
MAZ	Mazic Duro	Bis-GMA, TEGDMA, silica, Ba-glass, ZrO_2_	77 wt.%	DH6504A2	Vericom, Korea
DRA	Duro Ace	UDMA, Bis-EMA, silica, Ba-glass	85 wt.%	1002-2	

* Bis-MEPP, 2,2-bis(4-methacryloxypolyethoxyphenyl)propane; UDMA, urethane dimethacrylate; DMA, dimethacrylate; TEGDMA, triethylene glycol dimethacrylate; DEGDMA, diethylene glycol dimethacrylate; Bis = MPEPP, bisphenol A polyethoxy methacrylate, NPGDMA, neopentyl glycol dimethacrylate; Bis-GMA, bisphenol A diglycidylether methacrylate; Bis-EMA, ethoxylated bisphenol A dimethacrylate.

**Table 2 materials-13-04722-t002:** Color change (ΔE) of samples as a function of immersion periods in 3 different solutions.

Solution	Code	1 Week	2 Weeks	4 Weeks	6 Weeks	8 Weeks	12 Weeks
Deionized water	CER2	0.87 ± 0.67	1.04 ± 0.70	0.82 ± 0.24	1.13 ± 0.40	0.97 ± 0.68	1.66 ± 0.16
	CER3	0.58 ± 0.75	0.91 ± 0.65	0.74 ± 0.82	0.76 ± 0.49	0.52 ± 0.37	1.55 ± 0.45
	KZR	0.71 ± 0.15	0.79 ± 0.20	0.80 ± 0.13	0.94 ± 0.06	0.80 ± 0.21	0.98 ± 0.24
	KZR3	1.14 ± 0.36	1.13 ± 0.38	1.36 ± 0.34	1.57 ± 0.61	1.61 ± 0.36	1.57 ± 0.43
	EST	0.27 ± 0.15	0.48 ± 0.38	0.69 ± 0.27	1.03 ± 0.13	1.20 ± 0.28	2.04 ± 0.11
	ESTP	0.59 ± 0.29	0.53 ± 0.18	0.70 ± 0.30	0.50 ± 0.09	0.68 ± 0.28	0.85 ± 0.06
	AVE	0.53 ± 0.25	0.37 ± 0.08	0.34 ± 0.03	0.26 ± 0.03	0.29 ± 0.03	0.58 ± 0.01
	AVEP	2.23 ± 0.58	3.06 ± 0.76	3.67 ± 0.67	3.63 ± 0.56	3.76 ± 0.75	3.60 ± 0.75
	MAZ	1.07 ± 0.25	1.14 ± 0.25	1.14 ± 0.18	1.21 ± 0.20	1.18 ± 0.29	1.25 ± 0.27
	DRA	1.70 ± 0.30	1.82 ± 0.60	1.82 ± 0.37	1.63 ± 0.16	1.86 ± 0.29	2.15 ± 0.29
10% ethanol	CER2	0.62 ± 0.50	0.98 ± 0.01	0.45 ± 0.24	0.68 ± 0.34	0.77 ± 0.36	0.93 ± 0.39
	CER3	1.28 ± 0.13	1.24 ± 0.36	0.88 ± 0.22	1.33 ± 0.50	1.34 ± 0.55	1.52 ± 0.49
	KZR	1.03 ± 0.42	1.27 ± 0.35	1.04 ± 0.22	1.17 ± 0.39	0.97 ± 0.22	1.13 ± 0.38
	KZR3	1.05 ± 0.46	1.14 ± 0.15	1.40 ± 0.21	1.06 ± 0.49	1.63 ± 0.24	1.51 ± 0.75
	EST	0.38 ± 0.10	0.42 ± 0.04	0.56 ± 0.16	0.67 ± 0.17	0.73 ± 0.04	1.45 ± 0.23
	ESTP	0.31 ± 0.09	0.50 ± 0.21	0.45 ± 0.15	0.60 ± 0.13	0.52 ± 0.07	0.93 ± 0.19
	AVE	0.52 ± 0.14	0.62 ± 0.17	0.47 ± 0.15	0.48 ± 0.13	0.44 ± 0.15	0.58 ± 0.10
	AVEP	2.19 ± 0.04	3.07 ± 0.08	3.41 ± 0.07	3.46 ± 0.17	3.59 ± 0.10	3.51 ± 0.06
	MAZ	1.07 ± 0.12	1.26 ± 0.13	1.23 ± 0.19	1.18 ± 0.20	1.23 ± 0.14	1.38 ± 0.01
	DRA	1.54 ± 0.26	1.74 ± 0.21	1.62 ± 0.12	1.65 ± 0.12	1.48 ± 0.40	1.84 ± 0.38
simulated red wine	CER2	0.88 ± 0.03	0.90 ± 0.30	3.08 ± 0.47	4.90 ± 0.48	5.72 ± 0.75	7.16 ± 1.15
	CER3	1.23 ± 0.39	1.21 ± 0.44	4.52 ± 0.52	5.57 ± 0.94	7.05 ± 1.07	9.63 ± 1.09
	KZR	0.98 ± 0.39	1.15 ± 0.42	5.73 ± 0.54	5.74 ± 1.81	5.48 ± 1.48	5.58 ± 0.90
	KZR3	0.83 ± 0.34	0.93 ± 0.44	3.78 ± 1.08	5.59 ± 1.36	7.71 ± 1.74	9.59 ± 1.71
	EST	0.59 ± 0.40	0.67 ± 0.31	2.51 ± 0.65	3.18 ± 0.74	3.19 ± 0.49	4.52 ± 0.53
	ESTP	0.48 ± 0.09	0.55 ± 0.16	3.28 ± 0.40	3.95 ± 0.43	4.26 ± 0.52	4.45 ± 0.27
	AVE	0.47 ± 0.15	0.64 ± 0.27	0.65 ± 0.09	1.18 ± 0.20	1.43 ± 0.08	2.07 ± 0.25
	AVEP	1.85 ± 0.43	2.58 ± 0.43	5.60 ± 0.91	6.66 ± 0.75	7.55 ± 0.56	8.50 ± 0.81
	MAZ	0.97 ± 0.09	1.12 ± 0.12	2.26 ± 0.29	2.79 ± 0.27	3.66 ± 0.23	3.95 ± 0.29
	DRA	0.98 ± 0.20	1.32 ± 0.45	2.52 ± 0.43	2.66 ± 0.15	3.50 ± 0.19	3.51 ± 0.56

**Table 3 materials-13-04722-t003:** Translucency parameter change (ΔTP) of samples as a function of immersion periods in 3 different solutions.

Solution	Code	1 Week	2 Weeks	4 Weeks	6 Weeks	8 Weeks	12 Weeks
Deionized water	CER2	−0.11 ± 0.59	−0.40 ± 0.51	−0.25 ± 0.18	−0.45 ± 0.32	−0.19 ± 0.56	−0.12 ± 0.11
	CER3	−0.13 ± 0.48	−0.31 ± 0.45	−0.25 ± 0.46	−0.34 ± 0.34	−0.04 ± 0.21	−0.18 ± 0.23
	KZR	−0.37 ± 0.06	−0.32 ± 0.11	−0.27 ± 0.06	−0.38 ± 0.08	−0.27 ± 0.09	−0.10 ± 0.11
	KZR3	−0.49 ± 0.11	−0.47 ± 0.19	−0.45 ± 0.12	−0.43 ± 0.43	−0.63 ± 0.26	−0.27 ± 0.18
	EST	−0.06 ± 0.18	−0.17 ± 0.07	−0.11 ± 0.30	−0.15 ± 0.13	−0.15 ± 0.14	0.03 ± 0.04
	ESTP	−0.10 ± 0.23	−0.03 ± 0.29	0.02 ± 0.15	0.16 ± 0.24	0.03 ± 0.15	0.28 ± 0.21
	AVE	0.11 ± 0.08	0.11 ± 0.07	0.07 ± 0.05	0.12 ± 0.08	0.06 ± 0.01	0.24 ± 0.06
	AVEP	−0.94 ± 0.31	−1.33 ± 0.35	−1.45 ± 0.54	−1.75 ± 0.22	−1.68 ± 0.25	−1.36 ± 0.32
	MAZ	−0.63 ± 0.09	−0.48 ± 0.41	−0.51 ± 0.19	−0.44 ± 0.47	−0.46 ± 0.28	−0.19 ± 0.30
	DRA	−0.52 ± 0.08	−0.60 ± 0.19	−0.52 ± 0.10	−0.47 ± 0.10	−0.50 ± 0.18	−0.39 ± 0.17
10% ethanol	CER2	−0.17 ± 0.31	−0.38 ± 0.07	0.17 ± 0.15	0.03 ± 0.33	0.24 ± 0.25	0.05 ± 0.23
	CER3	−0.72 ± 0.34	−0.40 ± 0.40	−0.24 ± 0.14	−0.44 ± 0.50	−0.42 ± 0.50	−0.32 ± 0.65
	KZR	−0.37 ± 0.32	−0.44 ± 0.29	−0.27 ± 0.31	−0.37 ± 0.22	−0.37 ± 0.18	−0.16 ± 0.42
	KZR3	−0.55 ± 0.34	−0.51 ± 0.11	−0.42 ± 0.29	−0.41 ± 0.33	−0.56 ± 0.68	−0.13 ± 0.77
	EST	−0.12 ± 0.02	−0.11 ± 0.10	−0.11 ± 0.04	−0.14 ± 0.13	−0.08 ± 0.03	0.18 ± 0.11
	ESTP	−0.16 ± 0.02	−0.13 ± 0.09	−0.01 ± 0.06	0.06 ± 0.18	0.07 ± 0.15	0.06 ± 0.05
	AVE	0.09 ± 0.09	0.19 ± 0.07	0.17 ± 0.05	0.17 ± 0.09	0.16 ± 0.09	0.21 ± 0.09
	AVEP	−1.16 ± 0.10	−1.66 ± 0.20	−1.77 ± 0.11	−1.84 ± 0.18	−1.83 ± 0.18	−1.54 ± 0.12
	MAZ	−0.48 ± 0.18	−0.45 ± 0.16	−0.36 ± 0.31	−0.33 ± 0.21	−0.39 ± 0.15	−0.03 ± 0.23
	DRA	−0.64 ± 0.21	−0.62 ± 0.20	−0.60 ± 0.14	−0.53 ± 0.09	−0.46 ± 0.25	−0.24 ± 0.34
Simulated red wine	CER2	0.02 ± 0.11	0.28 ± 0.06	2.14 ± 0.79	3.13 ± 0.99	4.16 ± 1.00	5.73 ± 1.07
	CER3	−0.39 ± 0.23	−0.16 ± 0.28	1.95 ± 0.25	2.80 ± 0.28	3.50 ± 0.09	4.55 ± 0.26
	KZR	−0.06 ± 0.03	−0.07 ± 0.03	4.04 ± 0.26	4.51 ± 0.36	3.83 ± 0.45	3.77 ± 0.72
	KZR3	−0.18 ± 0.12	0.03 ± 0.29	2.94 ± 0.69	3.26 ± 0.81	4.61 ± 1.52	6.01 ± 1.75
	EST	−0.10 ± 0.07	0.04 ± 0.14	1.21 ± 0.16	1.84 ± 0.31	2.18 ± 0.09	2.09 ± 0.24
	ESTP	0.15 ± 0.10	0.22 ± 0.06	2.54 ± 0.20	3.13 ± 0.40	3.18 ± 0.24	3.25 ± 0.28
	AVE	−0.13 ± 0.16	−0.01 ± 0.08	0.59 ± 0.17	1.01 ± 0.12	1.13 ± 0.21	1.27 ± 0.16
	AVEP	−0.82 ± 0.10	−1.08 ± 0.06	−0.35 ± 0.26	0.22 ± 0.09	0.67 ± 0.47	1.77 ± 0.50
	MAZ	−0.27 ± 0.09	−0.25 ± 0.09	0.77 ± 0.14	1.84 ± 0.61	2.22 ± 0.26	3.27 ± 0.26
	DRA	−0.31 ± 0.04	−0.40 ± 0.14	0.16 ± 0.21	0.52 ± 0.21	0.58 ± 0.15	1.25 ± 0.26

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
