# Peer review of "Color Stability of Dental Reinforced CAD/CAM Hybrid Composite Blocks Compared to Regular Blocks"

_materials, 2020, doi:10.3390/ma13214722_

Round 1

Reviewer 1 Report

In this work, the author reports the color stability of reinforced CAD/CAM hybrid composite blocks compared to regular block. This manuscript needs major revision before publication. 

  1. Abstract should be more quantitative. Use full forms of (CER2/CER3, KZR/KZR3, EST/ESTP, AVE/AVEP, MAZ/DRA) in the abstract. Or deleted the sample code………..
  2. The introduction part should be more informative with recently published papers.
  3. Please provide the standard parameters for color measurement.
  4. If possible, please provide digital images of the contact angle experiment.
  5. Please check the transparency by using Uv-Vis spectroscopy.
  6. The author should provide mechanical properties (tensile and flexural test).

Author Response

  1. Abstract should be more quantitative. Use full forms of (CER2/CER3, KZR/KZR3, EST/ESTP, AVE/AVEP, MAZ/DRA) in the abstract. Or deleted the sample code………..

          We used full name instead of abbreviation.

  1. The introduction part should be more informative with recently published papers.

         We added new paragraphs.

  1. Please provide the standard parameters for color measurement.

        Parameters for color measurement results could not be added in the manuscript because of too many data.

  1. If possible, please provide digital images of the contact angle experiment.

        Not applicable

  1. Please check the transparency by using Uv-Vis spectroscopy.

        Translucency evaluation is mainly done using color coefficient measurement in dentistry. We will check using UV-Vis spectroscopy.

  1. The author should provide mechanical properties (tensile and flexural test).

        We added the unpublished data (flexural strength), which was tested in this lab.

Reviewer 2 Report

The paper presents the clinical esthetic stability of CAD/CAM hybrid composite block. It mainly talks about the color change and translucency parameter change of blocks in different solutions. It is a topic of interest to the researchers in the related areas and the paper may be accepted after minor modification. My detailed comments are as follows:

  1. The method used in the paper works very well to simulate the actual application environment. Compared with the regular blocks, the CAD/CAM hybrid composite blocks show excellent performance.
  2. For the above reason, the presentation calculated the parameters on the color stability of esthetic restorations. We can find the details through the table. However, some results are clinical unacceptable. The graph can show the news more intuitively than the table. The authors may show the better blocks through the paragraphs.
  3. For the figure 1, the description needs to be improved. For example, if the surface paragraphs are related to one parameter.
  4. Your paper needs careful editing by someone with expertise in technical English on English grammar, spelling, and sentence structure.

Author Response

  1. The method used in the paper works very well to simulate the actual application environment. Compared with the regular blocks, the CAD/CAM hybrid composite blocks show excellent performance.

  1. For the above reason, the presentation calculated the parameters on the color stability of esthetic restorations. We can find the details through the table. However, some results are clinical unacceptable. The graph can show the news more intuitively than the table. The authors may show the better blocks through the paragraphs.

         We added new figure.

  1. For the figure 1, the description needs to be improved. For example, if the surface paragraphs are related to one parameter.

        We tried to improve the caption.

Reviewer 3 Report

The authors studied the colour stability of reinforced CAD/CAM hybrid composite blocks to that of regular blocks. The work is interesting, and the following are my comments:

Title: the title may mislead materials readers. However, CAD/CAM are common in dentistry. They are common in engineering as the term of design and machining not really with composite materials. So suggest to modify the title and add density or dental and materials in the title.

L29: assisted to aided

Provide details of the composition of the abbreviated materials listed in table 1 like Bis-MEPP, etc.

Provide images if possible, to the colour changes of the same samples if they are visually distinguished.

Discuss the high range standard deviation of most of the sample.

Add better scale bar for the SEM images

The conclusion is brief and should include more key outcomes.

Author Response

The authors studied the colour stability of reinforced CAD/CAM hybrid composite blocks to that of regular blocks. The work is interesting, and the following are my comments:

Title: the title may mislead materials readers. However, CAD/CAM are common in dentistry. They are common in engineering as the term of design and machining not really with composite materials. So suggest to modify the title and add density or dental and materials in the title.

   We added “dental” to the title:

L29: assisted to aided

   We corrected it

Provide details of the composition of the abbreviated materials listed in table 1 like Bis-MEPP, etc.

   We added the full name of the abbreviations,

Bis-MEPP: 2,2-bis(4-methacryloxypolyethoxyphenyl)propane

UDMA: urethane dimethacrylate

DMA: dimethacrylate

TEGDMA: triethylene glycol dimethacrylate

DEGDMA: diethylene glycol dimethacrylate

Bis-MPEPP: bisphenol A polyethoxy methacrylate

NPGDMA: neopentyl glycol dimethacrylate

Bis-GMA: bisphenol A diglycidylether methacrylate

Bis-EMA: ethoxylated bisphenol-A dimethacrylate

Provide images if possible, to the colour changes of the same samples if they are visually distinguished.

   We added new Figure.

Discuss the high range standard deviation of most of the sample.

   The high standard deviation may be due to the formation of non-homogeneous discoloration in the specimen and the measuring spot (diameter, 6 mm) was not exact same place.

Add better scale bar for the SEM images

   We added new scale bar.

The conclusion is brief and should include more key outcomes.

   We added clinical significant suggestions.

Round 2

Reviewer 1 Report

The authors have addressed all the major concerns. So I recommend it for publication. 

Reviewer 2 Report

Accept